# Structure and Functions of Cocoons Constructed by Eri Silkworm

**DOI:** 10.3390/polym12112701

**Published:** 2020-11-16

**Authors:** Bin Zhou, Huiling Wang

**Affiliations:** 1College of Textile Science and Engineering (International Institute of Silk), Zhejiang Sci-Tech University, Hangzhou 310000, China; sanlin2007@126.com; 2School of Textiles and Clothing, Yancheng Polytechnic College, Yancheng 224005, China

**Keywords:** E cocoon, multilayer structure, functions, moisture buffer, temperature damping, anti-UV performance

## Abstract

Eri silkworm cocoons (E cocoons) are natural composite biopolymers formed by continuous twin silk filaments (fibroin) bonded by sericin. As a kind of wild species, E cocoons have characteristics different from those of *Bombyx mori* cocoons (B cocoons). E cocoons have an obvious multilayer (5–9 layers) structure with an eclosion hole at one end and several air gaps between the layers, which can be classified into three categories—cocoon coat, cocoon layer, and cocoon lining—with varying performance indexes. There is a significant secondary fracture phenomenon during the tensile process, which is attributed to the high modulus of the cocoon lining and its dense structure. Air gaps provide cocoons with distinct multistage moisture transmission processes, which form a good moisture buffer effect. Temperature change inside cocoons is evidently slower than that outside, which indicates that cocoons also have an obvious temperature damping capability. The eclosion hole does not have much effect on heat preservation of E cocoons. The high sericin content of the cocoon coat, as well as the excellent ultraviolet absorption and antimicrobial abilities of sericin, allows E cocoons to effectively prevent ultraviolet rays and microorganisms from invading pupae. The ultraviolet protection factor (UPF) of the E cocoon before and after degumming were found to be 17.8% and 9.7%, respectively, which were higher than those of the B cocoon (15.3% and 4.4%, respectively), indicating that sericin has a great impact on anti-UV performance. In the cocoon structure, the outer layer of the cocoon has 50% higher content than the inner layer, and the E cocoon shows stronger protection ability than the B cocoon. Understanding the relationship between the structure, property, and function of E cocoons will provide bioinspiration and methods for designing new composites.

## 1. Introduction

Cocoon is a type of unique and important biopolymer composite in nature with excellent microstructure and ecological functions. Cocoons of different shapes and structures are formed in a programmed manner through the regular swing of the head and the cyclical bending and stretching of the body to adapt to different environments [1]. The process of silk spinning and cocoon construction has undergone long-term natural selection and extensive evolution. Although cocoons are thin and lightweight, they can protect silkworms from various invasions in nature and provide a good place for silkworm metabolism [2,3,4]. Silk from *Bombyx mori* (mulberry silkworm) has been the most commonly used silk for centuries [5]. However, wild varieties, such as eri silkworms, have been partially domesticated for commercial applications [3,4,6]. Eri silkworms are the third largest silkworms in the world after *Bombyx mori* and tussah. They have only half the life cycle of *Bombyx mori* but construct bigger cocoons than *Bombyx mori* [2]. Eri silkworms are highly adaptable multifood silkworms that feed on leaves from several trees (castor leaves, cassava leaves, crane leaves, stinky leaves, pine leaves, cypress leaves, and so on) and are therefore easier to rear compared to *Bombyx mori*, which can be only fed on mulberry leaves [7,8,9]. Although the annual production of commercial E cocoons has reached tens of thousands of tons, there has not been much research on them. Due to the existence of eclosion hole and the loose structure of E cocoons, it is not possible to use traditional silk equipment to produce continuous filament to develop high-grade silk products, which limits their application [10]. Earlier studies have mainly focused on the breeding of eri silkworms as well as the biological structure, gene sequence, properties, and application of eri silk [1,11,12,13,14]. It has also been reported that the proteins generated by eri silkworms have antimicrobial activity [15]. Due to its excellent properties of moisturizing, antioxidation, and antiaging, silk protein has been widely used in nonwoven fields, such as cosmetics and biomedicine, in recent years [2,11].

Compared with *Bombyx mori*, eri silkworms live in an open and complex external environment, which requires a much higher level of protection [16]. It has been reported that silk fibroin and sericin derived from wild silkworms have better potential in the fields of medical and healthcare compared to *Bombyx mori* [7]. Limited research has recently been conducted on the performance and function of E cocoons. However, most studies take B cocoons as the research object, with attention paid to the relationship between the cocoon layer composition, structure, and mechanical properties of B cocoons [17,18,19,20]. Quantitative analysis and modeling have also been carried out to characterize the contribution of this structure to mechanical properties [21,22,23]. B cocoons have a nonobvious multilayer structure with hierarchical tensile mechanical properties from the outer layer to the inner layer [1,21]. The inner layer has finer silk and denser microstructure than other cocoon layers, so it has superior static and dynamic characteristics [18]. Stretching, compression, and gas diffusion properties of silkworm cocoons are closely related to the cocoon structure. Silkworm cocoons have interesting unidirectional gas transfer properties, which is beneficial for the survival of silkworm pupae [24,25,26]. Calcium oxalate crystals exist on the outer surface of cocoons, which can produce unique protective functions [26,27]. The special structure and components of the cocoon also provide it with excellent ability of temperature and humidity regulation and control [19,28]. Sericin, silk fibroin, and calcium oxalate in the cocoon layer have a certain capacity for adsorption of UVA, UVB, and UVC from sunlight, respectively [29].

The shape, size, structure, and properties of cocoons are different due to the diversity of genes, living environment, diet, and life cycle of silkworms, meaning cocoons have some unique characteristics. In this work, we studied the structural characteristics and differences in performance of every layer of the E cocoon as well as their role in the cocoon’s mechanical protection, humidity control, temperature buffering, and UV protection. A better understanding of the relationship between structure, property, and function of this important biological material will provide a conceptual platform to increase understanding of E cocoons and enlighten the design and production of protective nonwoven composites.

## 2. Materials and Methods

### 2.1. Materials

The E and B cocoons were supplied by Sericultural Research Institute, Chinese Academy of Agricultural Sciences (Zhenjiang, China). The chemicals used for the study were purchased from Shanghai Chemistry Reagent Co., Ltd. (Shanghai, China).

Preparation of cocoon materials: Experiments were performed on the cocoons before and after degumming or demineralization. Pupae were removed from the two kinds of cocoons, with some of the E cocoon divided into three parts according to outer (cocoon coat), middle (cocoon layer), and inner (cocoon lining) layers.

Demineralization: Partial cocoon materials were demineralized (removal of calcium oxalate monohydrate crystals) by treating them with ethylenediaminetetraacetic acid (EDTA) disodium salt solution (0.5 M) prepared in NaOH solution with the pH of the mixed solution kept at 7.5–8. Demineralization only removes calcium oxalate without affecting sericin [27].

Degumming: Cocoon materials were degummed by treating them in 0.5% Na_2_CO_3_ solution for 40–50 min at 80–90 °C with a solution to cocoon materials ratio of 30:1. All cocoon materials were washed carefully with deionized water and then dried at 60 °C [30,31]. We verified the absence of crystals and sericin by SEM imaging after demineralization and degumming processes, respectively.

### 2.2. Morphology

Appearance construction images of the cocoons were obtained using a digital imaging device. The microstructure was analyzed under a SU8010 SEM (Hitachi, Tokyo, Japan) at a voltage of 1 or 5 kV after gold-sputter coating.

### 2.3. Specifications and Mechanical Properties

The basic specifications of the cocoons were measured by digital vernier caliper, electronic balance, fiber fineness analyzer, and so on. Cocoons were spirally cut into strips with width of 5 mm and length of 80–100 mm along the direction of 20° from longitudinal to prepare the tensile test samples (Figure 1), thereby effectively avoiding the defect that the strip cannot be completely straightened due to the arc structure of the cocoon itself. Tensile testing instrument (YG065, Yantai, China) was used for tensile testing, and all tests were carried out at room temperature with gauge length of 50 mm and at a speed of 200 mm/min. The corresponding stress–strain curve was obtained by dividing the load and displacement by the cross-sectional areas and the gauge length of the specimens, respectively. The thickness of each cocoon shell was measured by fabric thickness meter (YG141, Nantong, China) with a presser foot area of 50 mm^2^ and pressure of 0.2 ± 0.0005 kPa.

### 2.4. Air and Moisture Permeability

The air permeability of cocoons before and after demineralization was tested on a YG (B)461D fabric air permeability tester (HONGDA, Nantong, China) according to ISO 9237 with a test area of 5 cm^2^ and a pressure difference of 100 Pa. The amount of air (L) per unit time (s) passing through cocoon per unit area (m^2^) at a certain pressure difference (100 Pa) on both sides of the cocoon was finally obtained. The specimen was placed as shown in the diagram (Figure 2).

The inner cavity of the cocoons was approximately an ellipsoid, and the lengths of three axes were denoted as a, b, c, respectively. The surface areas of the entire cocoon and the cocoon tested can be determined from the following equations, respectively:(1)$$S(m^{2})=\frac{4}{3}\pi \left(ab+bc+ac\right)$$
(2)$$S_{\mathrm{test}}(m^{2})=\frac{G_{\mathrm{test}}}{G}\frac{4}{3}\pi \left(ab+bc+ac\right)$$

The air permeability of the cocoons can be obtained from the equation below:(3)$$APR(mm/s)=\frac{APR_{\mathrm{test}}\times 5}{S_{\mathrm{test}}}=\frac{15APR_{\mathrm{test}}\times G}{4\pi \times G_{\mathrm{test}}\times \left(ab+bc+ac\right)}$$
where *S* is surface areas of the entire cocoon, *S*_test_ is effective surface areas of cocoon tested, *APR* is the true value of air permeability rate of cocoon, *APR*_test_ is the test value of air permeability rate of cocoon, *G* is the weight of the entire cocoon, and *G*_test_ is the weight of the cocoon that was tested on the instrument.

The moisture penetrability was measured using a YG601 fabric moisture tester (HONGDA, Nantong, China) based on GB/T 12704 (China) at specified conditions of 38 °C and 90% relative humidity with airflow velocity of 0.3 to 0.5 mm/s. The cocoon was fixed on a permeable cup containing a proper amount of dried moisture absorbent with effective test diameter of 2 cm considering the small size. The permeable cup was removed and weighed after 30, 60, 90, and 120 min. Moisture penetrability was calculated with the following Equation:(4)$$WVT({\mathrm{gm}}^{-2}h^{-1})=\frac{\Delta m(g)}{s(m^{2})\times t(h)}$$
where *WVT* is the moisture per unit area and unit time, Δ*m* is the weight difference of the permeable cup, *s* is the test area of the cocoon sample, and *t* is the experimental time.

Moisture absorption and desorption properties were implemented on an oven. For moisture desorption test, the cocoon materials were weighed every 15 min after placing 50 g of cocoons into an oven with the oven isothermal temperature setting at 70–80 °C. The test finished when the weight difference between adjacent results was less than 1%. For moisture absorption test, we recorded the weight of the cocoon after drying every 15 min in the standard environment of 20 °C and 65% relative humidity.

### 2.5. Thermal Behavior Test

Thermal property: Thermogravimetric analysis of cocoon materials was determined on a STA449 F3 thermogravimetric analyzer (NETZSCH, Selb, Germany) at a heating rate of 10 °C/min and a nitrogen flow rate of 20 mL/min.

Thermal insulation analysis: Temperatures were measured simultaneously both inside and outside the cocoons using two needle-type temperature probes (Figure 3). For E cocoon measurement, the temperature probe was placed into the cocoon through the eclosion hole end from which the moth escapes and the other end. Throughout the experiment, temperature change was divided into two stages: rising stage and falling stage. For the rising stage, a gradual rise of temperature was introduced by placing the cocoons into an oven with isothermal temperature setting at 50 °C; for the falling stage, a sudden fall of temperature was introduced by transferring the warmed cocoons that had nearly reached equilibrium from the oven to the air [19].

The thermal conductivity was measured on a TPS 2500 thermal constant analyzer (Hot disk, Stockholm, Sweden).

### 2.6. Anti-UV Performance Test

Ultraviolet protection factors (UPFs) of the cocoons before and after degumming were measured on a YG902C anti-UV and sunscreen protection tester (DARONG, Wenzhou, China) based on GB/T 18830 (China). According to the transmission of cocoons from sunlight ultraviolet radiation, the UVA transmittance T(UVA) in the wavelength range of 315–400 nm, the UVB transmittance T(UVB) in the wavelength range of 290–315 nm, and the UPFs were calculated according to the following equations:(5)$$T{\left(UVA\right)}_{i}=\frac{1}{m}\sum_{\lambda =315}^{400}T_{i}(\lambda )$$
(6)$$T{\left(UVB\right)}_{i}=\frac{1}{k}\sum_{\lambda =290}^{315}T_{i}(\lambda )$$
(7)$$UPF_{i}=\frac{\sum_{\lambda =290}^{\lambda =400}E(\lambda )\times \varepsilon (\lambda )\times \Delta \lambda }{\sum_{\lambda =290}^{\lambda =400}E(\lambda )\times T_{i}(\lambda )\varepsilon (\lambda )\times \Delta \lambda }$$
where *T_i_*(*λ*) is the spectral transmittance of the specimen *i* at a wavelength of *λ*; *m* and *k* are the number of measurements between 315–400 and 290–315 nm, respectively. *E*(*λ*) is solar spectral irradiance in units of W·m^−2^·nm^−1^, *ε*(*λ*) is the relative erythema effect, and Δ*λ* is the wavelength interval in nm.

### 2.7. Antibacterial Activity Test

The antimicrobial activity of the cocoon was tested against *Escherichia coli* and *Staphylococcus aureus* using a shaking flask method according to GB/T 20944.1 (China). *E. coli* dh5α and *S. aureus* were cultured in Luria–Bertani (LB) medium (220 rpm, 37 °C). Bacteria solution with a cell concentration of 1 × 10^8^ cfu/mL and OD600 of 0.6 was mixed with melted LB solid medium at a volume ratio of 1:1000. Then, 10 mL of mixed culture medium was poured into each petri dish (90 mm in diameter). A picture was taken after cocoon samples were added to the petri dish for 24 h at 37 °C.

### 2.8. Statistical Analysis

All the statistical data are expressed as the means ± standard deviation (SD). Statistical analyses were performed by SPSS 13.0, and differences were considered as significant for *p*-value < 0.05.

## 3. Results

### 3.1. Construction and Morphology of Cocoons

Figure 4a,b shows the appearance of an intact E cocoon as a spindle or jujube kernel. Although the external shapes of the E cocoon were not uniform, Figure 4c shows that the shape of cavity of the cocoon was regular and similar to an elliptical shape after stripping the cocoon coat and partial cocoon layers, and the end of the eclosion hole was radiative like a crater. The average diameter of the eclosion hole was about 2–3 mm with its vicinity covered with fluffy silk; the eclosion hole showed irregular polygona because of the folding cocoon lining, which might help the moth to drill out of the cocoon.

The specifications and properties of the E cocoon used in this research are shown in Table 1. As can be seen, the size and thickness of the E cocoon were much higher than those of the B cocoon, but the weight of the two cocoons was about the same. The weight per square meter of the E cocoon was lower than that of the B cocoon, which indicates that the E cocoon has a relatively loose structure.

The E cocoon could be divided into 5–9 layers, forming three parts called cocoon coat, cocoon layer, and cocoon lining, which had considerable variation in properties, as can be seen from Table 2. The thick and loose cocoon coat bulging like a fin on one side of the cocoon accounted for approximately 21% of the total weight of the cocoon. The cocoon coat and the cocoon layer had no obvious boundary, and there were a large number of discontinuous static air gaps between them, as can be seen from Figure 4d, which provides the cocoon with excellent thermal insulation and impact resistance [6,23]. The cocoon layer was in loose contact with the cocoon coat but in very tight contact with the cocoon lining, which accounted for about 71% of the weight of the cocoon and had a relatively compact structure, providing the most protection for larvae. The cocoon lining was paper-like with dense structure, which accounted for approximately 10% of the total weight of the cocoon. The cocoon lining actually constituted a “closed” cavity supporting larvae activity. The cocoon coat had high sericin content and moisture regain, but the cocoon layer had high strength and low elongation.

SEM images of the E cocoon are shown in Figure 5. As can be seen from Figure 5a, the silk on the outer surface of the cocoon coat was entangled with each other irregularly. There were various forms, such as twisting, folding, and interlacing, and the structure was relatively loose. Each silk was formed by two parallelly arranged monofilaments held together by sericin; the sericin was coated on the periphery of the silk fibroin for protection and acted as a glue between the silk cocoons when forming a cocoon layer. Some microfibrils of about 1 µm in diameter were embedded in the silk fibroin (indicated by arrows in Figure 5a). There were many irregularly sized and unevenly distributed blocks on the outer surface. Studies have shown that these blocks are calcium oxalate crystals with a side length of about 1–3 μm [27]. Its existence is not only conducive to improving the hardness of cocoons and absorbing ultraviolet C but also improving thermal stability by blocking still air in the cocoon [29]. The silk arrangement of the cocoon layer (Figure 5b) was more regular than that of the cocoon coat. The silk sericin distribution was uniform, and the content of calcium oxalate crystal was less than that in the cocoon coat. The surface of the cocoon lining (Figure 5c) was compact and smooth. Silk was woven together in “Y” type, “+” type, and cross knitted at different angles, and the number of calcium oxalate crystals was significantly less than that of the cocoon coat surface. It is often difficult for the sericin in the inner cocoon silk to coat all the silk fibroin (indicated by arrows in Figure 5c). When part of the cocoon coat near the eclosion hole was peeled off (Figure 5d), the fibers around the hole were thin and loose, and it was obvious that the silk turned at the hole. The distribution of calcium oxalate crystals on the fiber surface of the hole was less than that of other cocoon layers.

### 3.2. Tensile Properties of Cocoon

Stress–strain curves of rectangular specimens of E cocoon (100 mm × 50 mm × 1.12 mm) and B cocoon (100 mm × 50 mm × 0.39 mm) obtained from tensile tests are given in Figure 6. As can be seen, the stress–strain curves had three distinct regions: hook zone, yield zone, and fracture zone. The average Young’s modulus and stress of the E cocoon were 329.10 ± 66.51 and 44.53 ± 25.24 MPa, respectively, which were evidently lower than those of *Bombyx mori* (738.37 ± 172.46 MPa and 71.53 ± 20.18 MPa, respectively), while the average elongation of the E cocoon was 5.26 ± 1.37%, which was about 1.7 times that of *Bombyx mori* (3.10 ± 0.63%). The high modulus of the cocoon lining and its dense structure led to obvious fracture time differences in the tensile process. There were obvious secondary fracture phenomena on the stress–strain curve. The stress of the cocoon lining was about 1.5 and 1.8 times that of the cocoon layer and cocoon coat, respectively. Such greatly enhanced mechanical properties in the cocoon lining are important for a cocoon to efficiently protect the pupa.

The average bursting strength and displacement of the E cocoon were about 570 ± 11 N and 27.7 ± 2 mm, respectively, which were about 1.3 and 1.2 times those of B cocoon (475 ± 51 N and 23.1 ± 2 mm), respectively, indicating its high energy absorption capacity. This is attributed to the anisotropic distribution of the silk orientations resulting from the manner in which the silkworm spins silks.

### 3.3. Air and Moisture Permeability of Cocoon

Cocoons achieve good temperature stability by controlling static air. The air permeability reflects the ability of cocoons to control static air to a certain extent. Within a certain range, the smaller the air permeability of the nonwoven material, the better the heat retention.

The air permeability of the E cocoon layer and the eclosion hole before demineralization were 63.698 and 270.420 mm/s, respectively (Figure 7). The air permeability of the cocoon layer and the eclosion hole after demineralization were up to 91.286 and 293.765 mm/s, respectively, which indicates that calcium oxalate crystals on the surface of fibers can affect their control ability. The presence of calcium oxalate crystals helps to reduce the penetration of external air into cocoons. The air permeability of the B cocoon before and after demineralization were 125.6 and 141.289 mm/s, respectively, which were higher than those of the E cocoon layer and lower than those of the eclosion hole. The change in air permeability of the B cocoon before and after demineralization was less than that of the E cocoon, which verifies that the content of calcium oxalate in the E cocoon is larger than that of the B cocoon [27].

It can be seen from Figure 8 that in the initial stage (0–30 min), the moisture permeability of both cocoons was relatively low, and there was humidity difference on both sides of the cocoon layer. Water drops first wetted the surface and penetrated into the pores. The condensed liquid water was then transported to the other side of the cocoon by capillary effect (Figure 9) and evaporated into water vapor, which was absorbed by the moisture absorbent.

After this period, the moisture permeability tended to stabilize, with the moisture permeability of the E and the B cocoons being 101 and 122 (g/m^2^·h), respectively. Due to the multilayer structure of the E cocoon and the obvious air gaps between layers, there will be multiple processes of “wetting–permeation–evaporation–adsorption”, which form a good buffer effect.

The moisture absorption and desorption curves of the cocoon in the standard state are shown in Figure 10. The balance moisture regain of absorption of the E and B cocoons were 12.7% and 10.2%, respectively, while the balance moisture regain of desorption were 13.7% and 11.0%, respectively. There was obvious absorption hysteresis, and the balance moisture regain differences caused by absorption hysteresis were about 1% and 0.8%, respectively. These values are close to cotton fiber (0.9%) and less than wool (2%). Materials with high moisture regain will have high balance moisture regain difference. Under conditions of same moisture regain, the smaller the balance moisture regain difference, the higher the absorption and desorption rate will be [32]. The moisture regain of the E cocoon was higher than that of the B cocoon, but the absorption and desorption rates were smaller than that of the B cocoon, which coincides with the moisture permeability of the cocoon. The moisture absorption of the cocoon itself makes the cocoon layer area maintain the balance of “moisture absorption and desorption”.

### 3.4. Thermal Behavior Analysis

Figure 11 shows the temperature variation and temperature lag curves of locations inside and outside the cocoons when they were subjected to the process of temperature rise and fall, respectively. The temperature change inside the cocoon was slower than that outside the cocoon, indicating that the cocoon has obvious temperature buffering capability, which might be related to the natural habitat conditions that silkworms are required to defend against.

When the temperature rose and fell, the temperature of the E cocoon took longer to reach the target temperature than the B cocoon through two different probe insertion positions of eclosion hole end and noneclosion hole end, with lags of about 3 and 2 min, respectively. When the temperature rose, the maximum temperature lags were 7.7 and 5.4 °C, respectively. When the temperature fell, the maximum lag temperatures were 11.6 and 8.1 °C, respectively. The E cocoon exhibited better thermal damping ability than the B cocoon. The multilayer structure of the E cocoon, the air gaps existing between the cocoon layers, and the volume of the internal cocoon space all contribute to the thermal damping behavior of E cocoons. The thermal diffusion coefficient of the E cocoon was 0.2413 mm^2^/s, which was lower than that of the B cocoon (0.4567 mm^2^/s) and consistent with the results of air permeability of the two materials. The presence of calcium oxalate crystals helps to reduce the penetration of external air into the cocoon and blocks still air in the cocoon shell to enhance the thermal stability and affect the thermal behavior of the cocoon.

For the E cocoon, the test was performed from the eclosion hole and the noneclosion hole. The results were different but not obvious. When the temperature rose and fell, the time difference for reaching the target temperature was only 1 min. When the temperature rose, the maximum temperature lags inside and outside the cocoons were 7.7 and 7.4 °C, respectively. When the temperature fell, the maximum temperature lags inside and outside the cocoons were 11.6 and 10.9 °C, respectively. This shows that although the cocoon layer at the eclosion hole is not continuous, the silk distributed around the periphery densely “covers” the eclosion hole without affecting the overall temperature control ability of the cocoon. The thickness of the cocoon shell of the E cocoon was 1.12 ± 0.22 mm, and the weight per square meter was only 202.71 ± 16.38 g, which further proves that the E cocoon has a loose structure and also has a better thermal insulation effect.

The TG curves (Figure 12) show that the E cocoon, B cocoon, eri silk, and *Bombyx mori* silk after degumming had similar thermal decomposition behavior. The dehydration weight loss temperatures due to the moisture content of silk appeared at 115, 116, 123 and 107 °C, respectively. The weight loss rates of water molecules were 2.7%, 6.3%, 5.1% and 3.9%, respectively. With the temperature continuing to increase, the weight loss was relatively slow. The rapid weight loss temperature ranges were 220–431, 231–454, 227–409 and 238–399 °C, respectively. The rapid decomposition interval of degumming *Bombyx mori* silk was small. The weight loss rates of E cocoon, B cocoon, eri silk, and *Bombyx mori* silk after degumming were 61%, 72%, 66% and 64%, respectively. The weight loss rate of the E cocoon was lower than that of the B cocoon, which should be related to the higher sericin content of the B cocoon and shows that he E cocoon has better thermal stability. The weight loss rate of eri silk was close to that of *Bombyx mori* silk after degumming.

### 3.5. Anti-UV Performance Analysis

The wavelength of UV light in sunlight is between 200 and 400 nm, which is further subdivided into three main wavelength ranges, called UVA, UVB, and UVC, with wavelength ranges of 315–400, 290–315 and 200–290 nm, respectively. UVC is prevented from reaching the earth by the ozone layer in the atmosphere, while UVA and UVB can reach the surface of the earth in an amount sufficient to cause damage to living organisms. Figure 13 shows the anti UV performance of two kinds of cocoons. From calculations according to Equations (4)–(6), it was found that the UPF of the E cocoon before degumming was 17.8, which was higher than that of the B cocoon (15.3). The UVA and UVB transmittances, i.e., T(UVA) and T(UVB), were 7.4% and 3.58%, respectively, which were lower than those of the B cocoon (13.33% and 3.42%, respectively). The E cocoon exhibited better UV resistance than the B cocoon, which is also a necessary protection requirement for wild silkworm.

For the E cocoon, the contents of UPF, T(UVA) and T(UVB) after degumming were 9.70%, 12.00% and 7.26%, respectively, while for B cocoon, the contents were 4.40%, 25.37% and 15.97%, respectively, indicating that sericin has a great impact on the anti-UV performance of cocoons. Studies have shown that tyrosine, tryptophan, and phenylalanine in sericin can absorb UV effectively [1]. It is also a reasonable design of the silk cocoon structure that the outer layer of cocoons has 50% higher content than the inner layer. The sericin content of the outer layer of the B cocoon (37.42 ± 2.78) was much higher than that of the E cocoon (13.68 ± 0.49), but the UPF was lower than that of the E cocoon. The reason is that UV transmission is also related to the thickness, structure, and color depth of the material. The E cocoon has a thickness larger than that of B cocoon, has obvious multilayer structure, and is yellowish in color, so its anti-UV effect is better than that of B cocoon.

### 3.6. Antibacterial Activity Analysis

The experimental results (Figure 14) of antimicrobial activities showed that the diameters of the bacteriostatic circles of the E and B cocoons against *S. aureus* were 16.3 and 16.5 mm, respectively. The diameters of bacteriostatic circles against *E. coli* were 18.6 and 19.6 mm, respectively. This showed that the cocoons had bacteriostatic effect against both *S. aureus* and *E. coli*, but the antibacterial effect against *E. coli* was more significant than that against *S. aureus*. The two kinds of cocoons had similar bacteriostatic effect against *S. aureus*, while the antibacterial effect of B cocoon against *E. coli* was slightly stronger than that of the E cocoon. The antibacterial properties of silkworm cocoons can help protect them from microorganisms and ensure the pupae complete the development process smoothly.

## 4. Conclusions

E cocoons are natural composite polymer materials with a unique structure and function. In this study, we investigated the multilayer structure and functions of E cocoons, such as mechanical protection, temperature and humidity regulation, air and moisture permeability, UV protection, and the antimicroorganism needed for silkworm pupa metamorphosis.

The difference in structure and properties between the cocoon layers make the mechanical properties of E cocoons different from that of B cocoons. E cocoons have obvious secondary fracture phenomenon. The air gaps, content, and distribution of sericin and calcium oxalate provide E cocoons with better thermal insulation and temperature–humidity buffer effect. The high sericin content of the cocoon coat and its uniform distribution provide the cocoon with better antiultraviolet and antimicrobial characteristics.

Understanding the reasonable and quantitative structural model provided by E cocoons will help design or optimize multifunctional nonwoven products and will help expand the application field of cocoons.

## Figures and Tables

**Figure 1 polymers-12-02701-f001:**
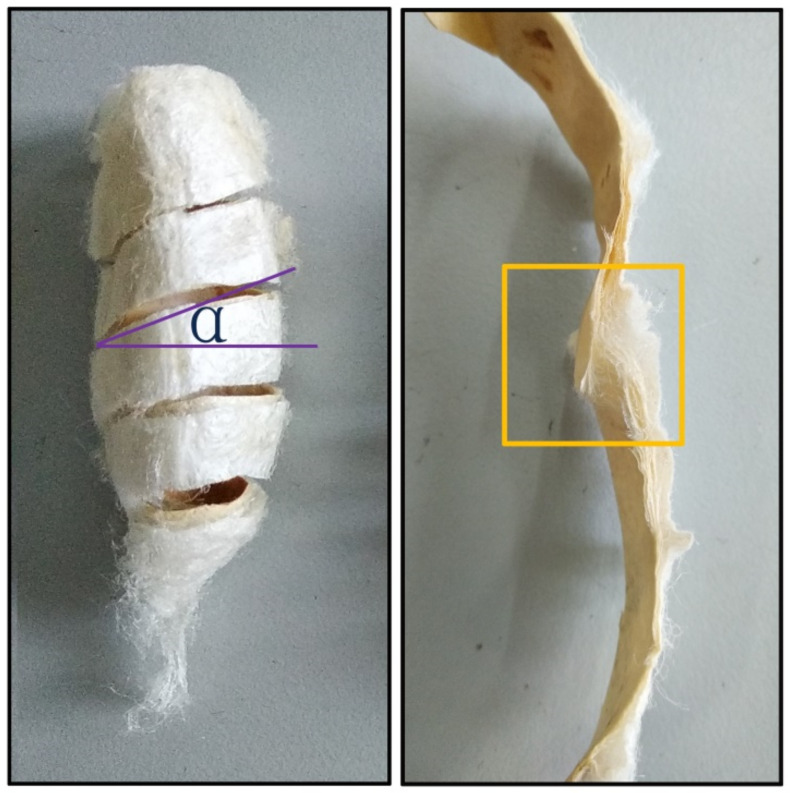
Schematic diagram of spiral sampling and digital images of fracture surfaces of cocoon shells after tensile failure. The yellow box shows the shape of the broken cocoon shell.

**Figure 2 polymers-12-02701-f002:**
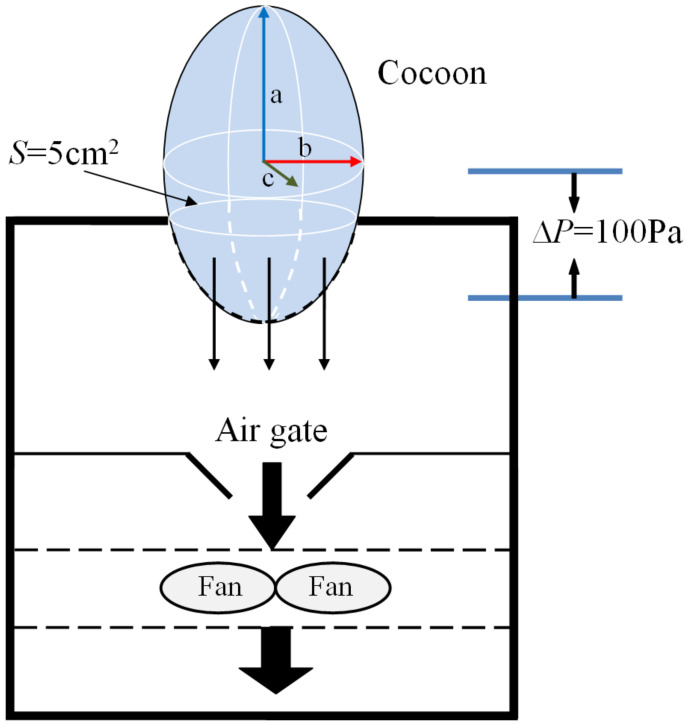
Schematic diagram of air permeability test of cocoon.

**Figure 3 polymers-12-02701-f003:**
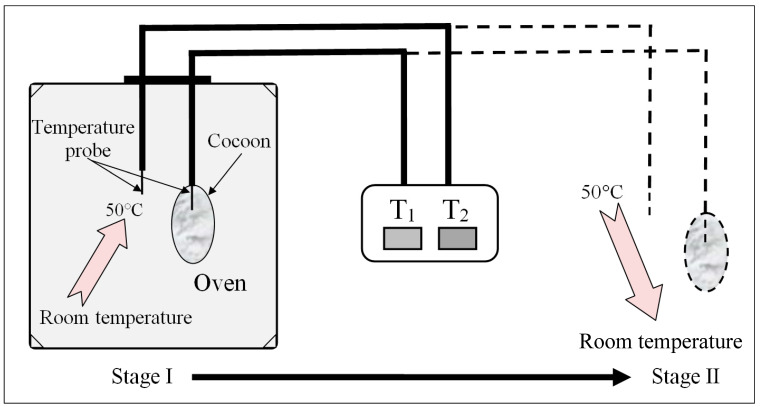
Schematic diagram of self-made cocoon temperature insulation performance test device and its test process.

**Figure 4 polymers-12-02701-f004:**
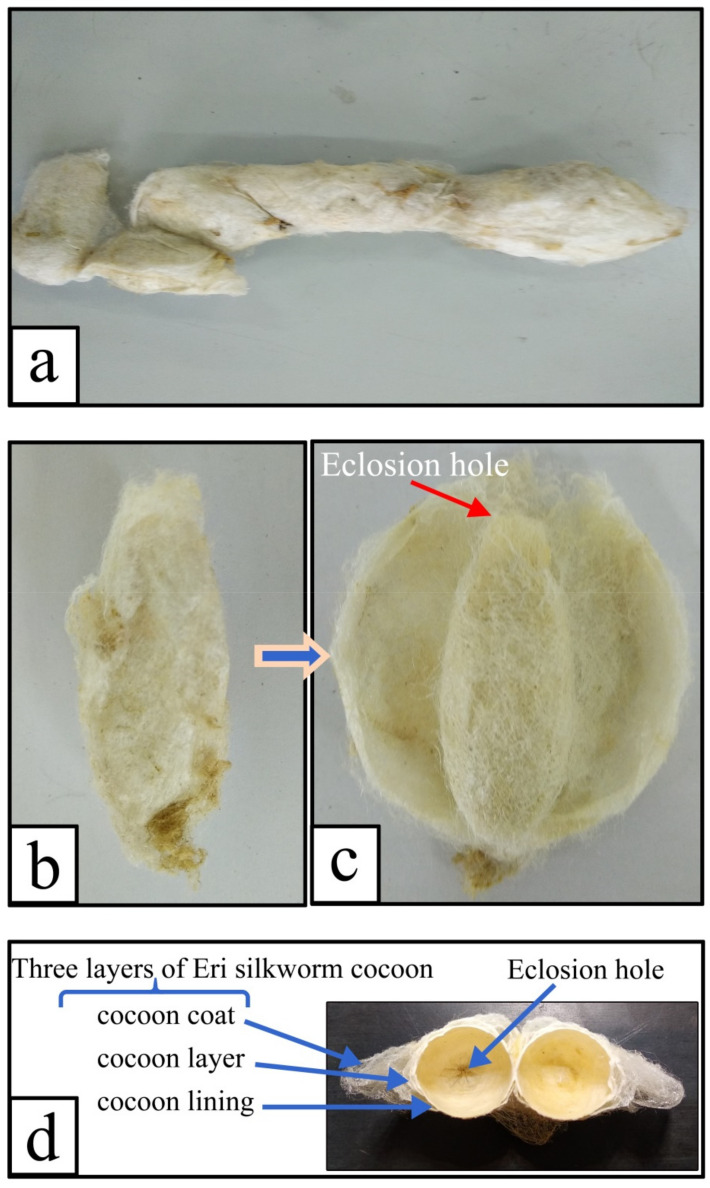
Digital images of the E cocoon: (**a**) diversity of E cocoon morphology; (**b**) intact E cocoon; (**c**) cocoon layer after the outer most layer had been cut open; (**d**) three layers (cocoon coat, cocoon layer, and cocoon lining) and eclosion hole of the cocoon.

**Figure 5 polymers-12-02701-f005:**
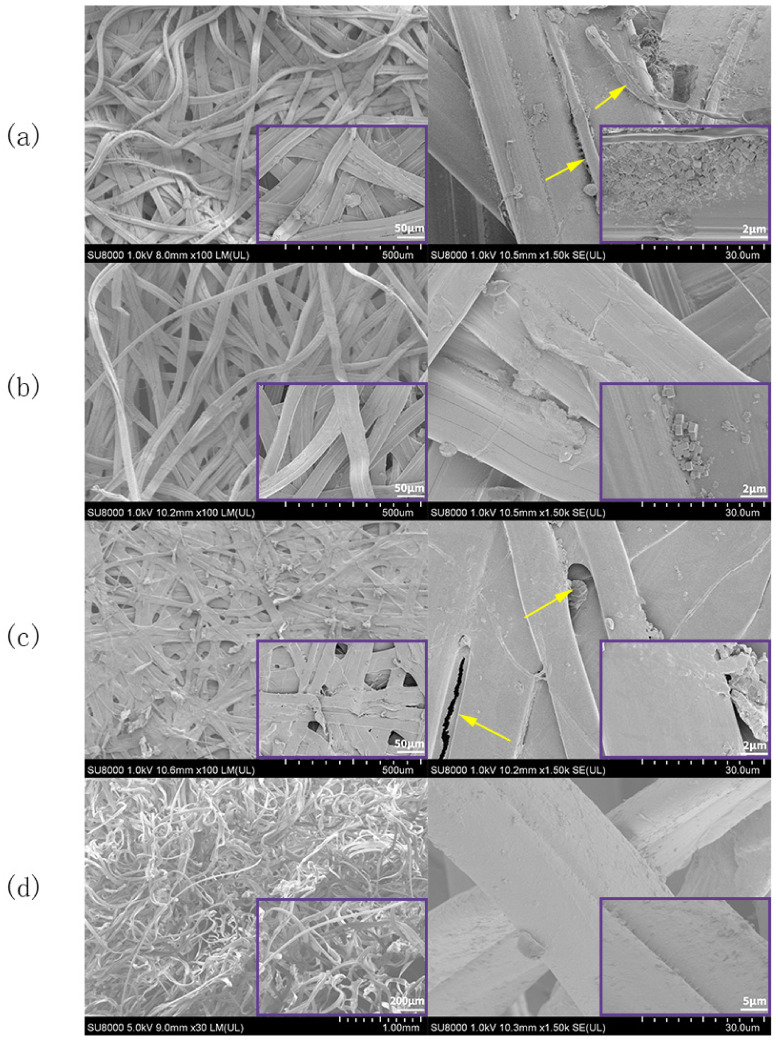
SEM images of the microstructures of E cocoon: (**a**) cocoon coat; (**b**) cocoon layer; (**c**) cocoon lining; and (**d**) outer layer of eclosion hole. The yellow arrow in Figure 5a shows microfabrils of about 1 μm in diameter. The yellow arrow in Figure 5c shows that sericin can not completely coat all silk fiber, The purple square frame (bottom right) shows a partial enlarged view.

**Figure 6 polymers-12-02701-f006:**
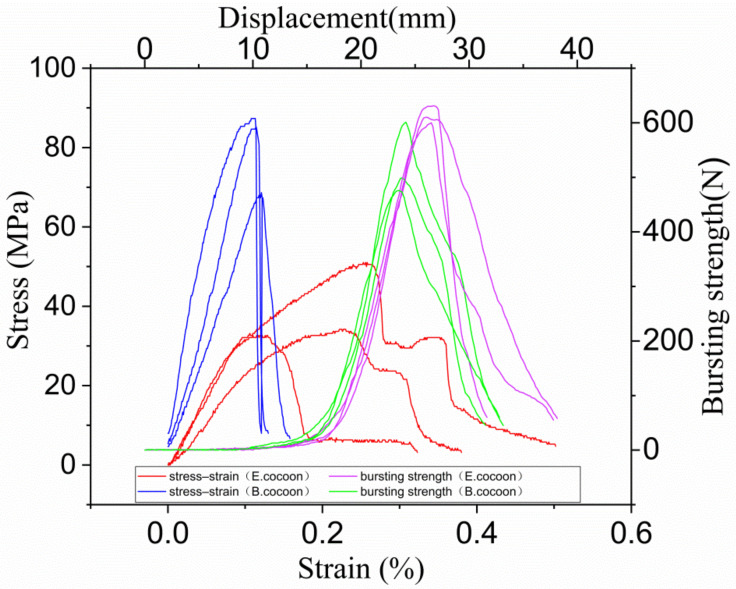
Stress–strain curves (**left**) and bursting strength curves (**right**) of the E and B cocoons.

**Figure 7 polymers-12-02701-f007:**
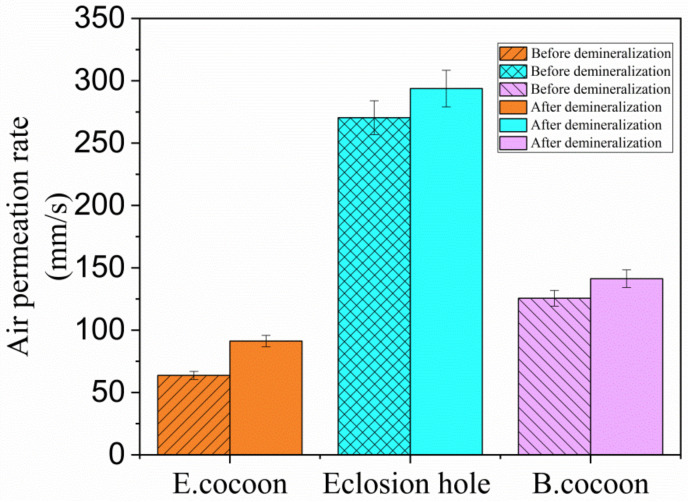
Air permeability rate values of E and B cocoons before and after demineralization.

**Figure 8 polymers-12-02701-f008:**
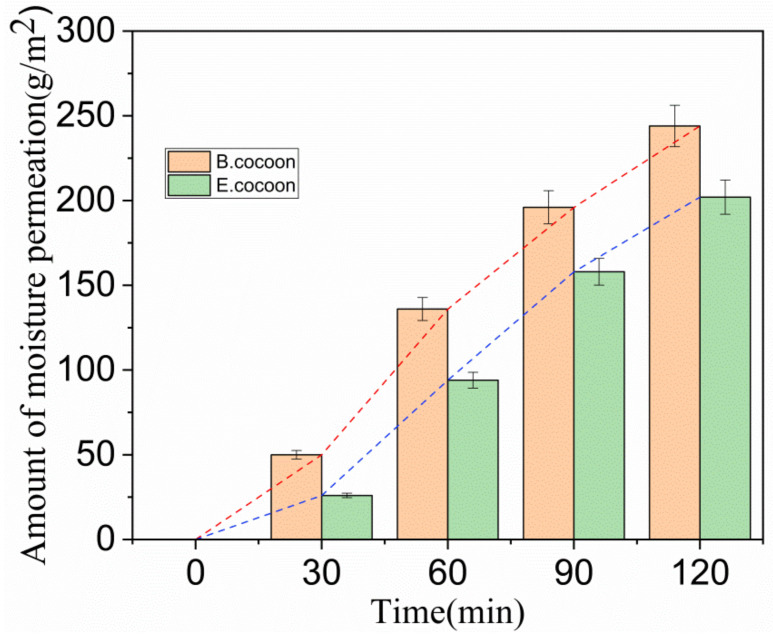
Relationship between amount of moisture permeability and time.

**Figure 9 polymers-12-02701-f009:**
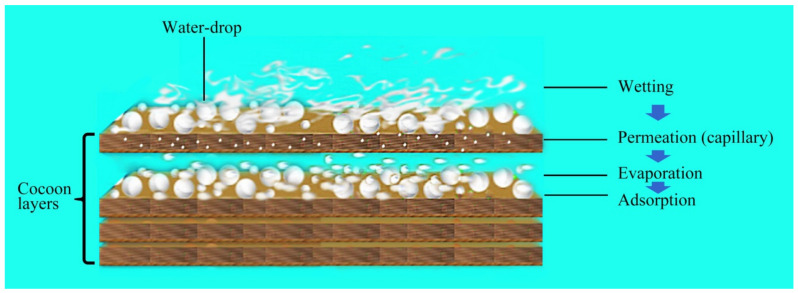
The multiple processes of moisture permeability of E cocoon.

**Figure 10 polymers-12-02701-f010:**
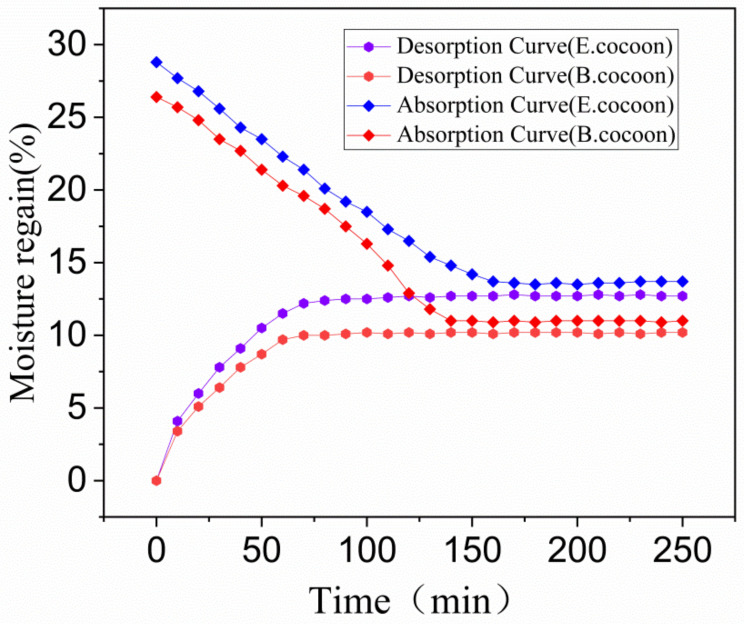
Moisture absorption and desorption curves of E and B cocoons.

**Figure 11 polymers-12-02701-f011:**
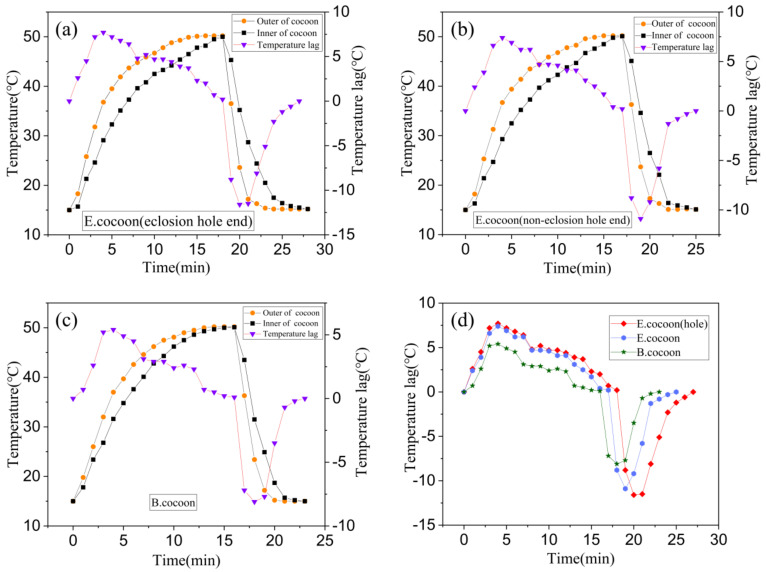
Temperature variation and temperature lag curves for locations both inside and outside the E and B cocoons. (**a**) The temperature probe was inserted into the cocoon from the eclosion hole of the E cocoon for testing; (**b**) the temperature probe was inserted into the cocoon from the noneclosion hole of the E cocoon for testing; (**c**) the temperature probe was inserted into the B cocoon for testing; (**d**) the temperature lag comparison curve.

**Figure 12 polymers-12-02701-f012:**
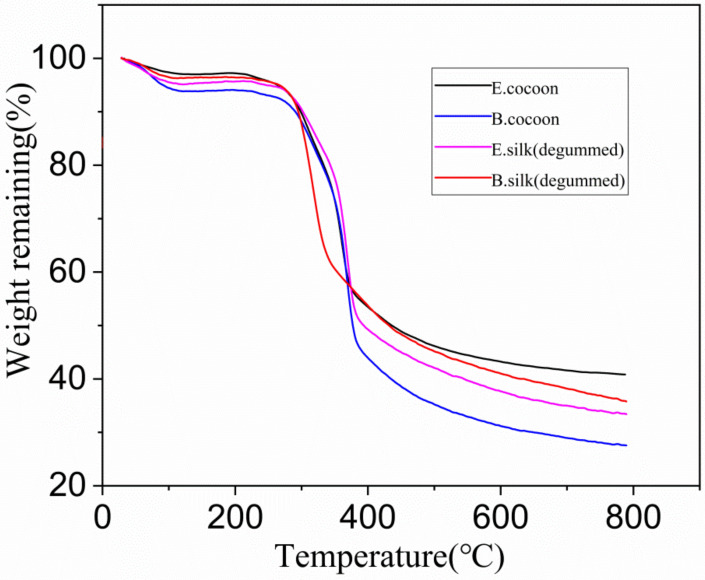
Thermogravimetric (TG) curves of cocoon shells and degummed fibers produced by E and B cocoons.

**Figure 13 polymers-12-02701-f013:**
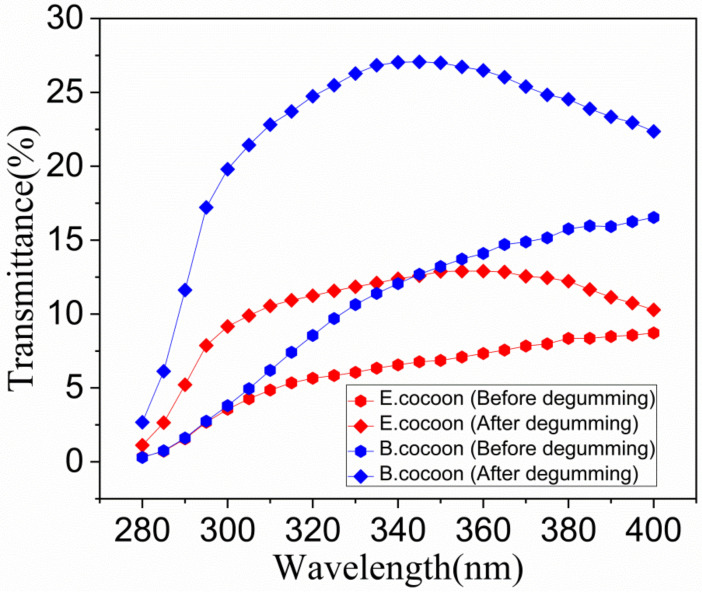
The anti-UV performance of the E and B cocoons before and after degumming.

**Figure 14 polymers-12-02701-f014:**
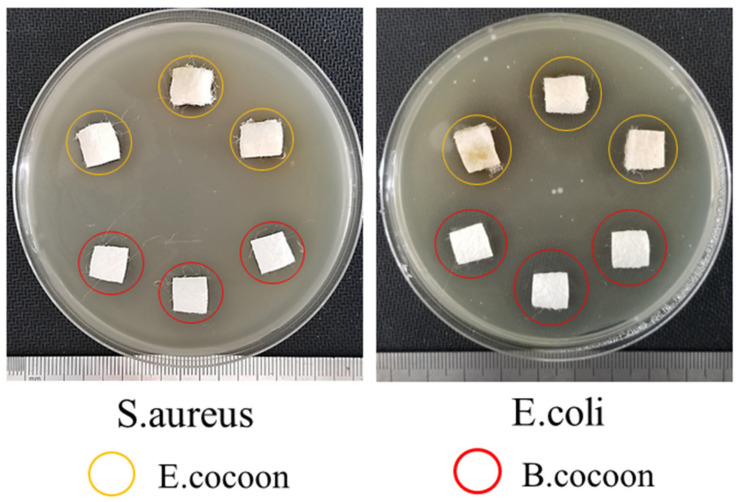
Antimicrobial activities of E and B cocoons.

**Table 1 polymers-12-02701-t001:** Geometrical parameters of the E and B cocoons.

Cocoon Type	Weight (g)	Weight per Square Meter (g/m^2^)	Thickness of Cocoon Shell (mm)	Longer Diameter of Cocoon (mm)	Shorter Diameter of Cocoon (mm)
E cocoon	0.38 ± 0.07	202.71 ± 16.38	1.12 ± 0.22	67.27 ± 0.43	41.13 ± 0.19 22.44 ± 0.19 ^a^
B cocoon	0.30 ± 0.02	289.57 ± 36.43	0.39 ± 0.02	31.57 ± 0.19	19.01 ± 0.17

^a^ The cross section of the E cocoon was irregular, so both the longest diameter and the shortest diameter were measured.

**Table 2 polymers-12-02701-t002:** Fiber properties of the three layers in E cocoon.

Performance Index	Cocoon Coat	Cocoon Layer	Cocoon Lining	Intact Cocoon
Thickness (mm)	0.29 ± 0.03	0.49 ± 0.05	0.37 ± 0.04	1.12 ± 0.22
Weight (g)	0.08 ± 0.14	0.27 ± 0.05	0.04 ± 0.01	0.38 ± 0.07
Fiber Fineness (den)	2.13 ± 0.15	2.75 ± 0.29	1.86 ± 0.14	2.63 ± 0.34
	1.94 ± 0.15 *	2.35 ± 0.25 *	1.77 ± 0.13 *	2.27 ± 0.17 *
Sericin Content (%)	13.68 ± 0.49	9.64 ± 0.47	5.95 ± 0.30	10.57 ± 0.43
Moisture Regain (%)	13.16 ± 0.32	12.34 ± 0.31	11.94 ± 0.22	12.71 ± 0.16
	12.52 ± 0.62 *	11.94 ± 0.42 *	11.17 ± 0.74 *	12.14 ± 0.27 *

* There were significant difference in some properties of E cocoon before and after degumming, so the fiber fineness and moisture regain after degumming were also measured.

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
