# Peer review of "Structure and Functions of Cocoons Constructed by Eri Silkworm"

_polymers, 2020, doi:10.3390/polym12112701_

Round 1

Reviewer 1 Report

The presented work focuses on the assessment of the structure and properties of the composite structure of cocoons form Eri Silkworms. In my opinion, the subject of the paper is very interesting, it takes up a very important subject of the properties of composite materials of natural origin.While the application potential of this type of materials is very limited, the presented work allows for a better understanding of the properties of natural composite, which in the future will allow to recreate some important features impossible to obtained by currently produced composites.

In my opinion, the presented manuscript is suitable for publication in its current form.

Author Response

Thank you for your kind suggestions and  comments for revising our paper (polymers-993884) titled “Structure and functions of cocoons constructed by Eri Silkworm” We have revised the manuscript accordingly, and all suggestions have been taken into full consideration.I have optimized the expression of the experimental results in this paper to make it more concise and clear.Enclosed please find the revised manuscript and responses to the reviewers. We sincerely hope our manuscript will be acceptable for publication in polymers.

Reviewer 2 Report

In this study, the authors presented the morphological and mechanical properties of Eri silkworm silk. In these days, the wild type silkworm silks are garnered more attention in the field of bio-based fibers, compared with the domesticated silk fibers. This is because the wild type silk fibers have a variety of excellent physical properties in terms of thermal and mechanical properties, as the authors described in the Introduction section in the present manuscript. Although the research topic is very interesting for this reviewer, the English quality of this manuscript is too bad. This reviewer strongly requests the authors that English correction service must be required for this manuscript before submission. Also, double-checking the manuscript is essential before submission.

1) Bombyx mori and Eri should be written in Italic throughout the manuscript.

2) In page 3, Figure 5 is first appeared in the manuscript. This is strange. Put Figures in an appropriate place in the manuscript.

3) The font sizes in all Figures are not consistent. The authors should be more careful to prepare each Figure.

4) In Figure 9, this reviewer cannot read the description due to the unclear letters in the Figure.

4) Make clear what the purple color squares denote at the caption in Figure 4.

5) Figure 12, Y axis should not be “Weigh residue” but “Weight remaining”.

6) In line 25 in page 7, 1 um should be 1 µm.

Author Response

Thank you for your kind suggestions and comments for revising our paper (polymers-993884) titled “Structure and functions of cocoons constructed by Eri Silkworm” We have revised the manuscript accordingly, and all suggestions have been taken into full consideration. Enclosed please find the revised manuscript. We sincerely hope our manuscript will be acceptable for publication in polymers.

Response to Reviewer’s Comments

Comments and Suggestions for Authors

In this study, the authors presented the morphological and mechanical properties of Eri silkworm silk. In these days, the wild type silkworm silks are garnered more attention in the field of bio-based fibers, compared with the domesticated silk fibers. This is because the wild type silk fibers have a variety of excellent physical properties in terms of thermal and mechanical properties, as the authors described in the Introduction section in the present manuscript. Although the research topic is very interesting for this reviewer, the English quality of this manuscript is too bad. This reviewer strongly requests the authors that English correction service must be required for this manuscript before submission. Also, double-checking the manuscript is essential before submission.

Replying:

Thanks for the reviewer’s kind suggestion. we have commissioned a special English embellishment agency to adjust the article, and the detailed revised contents are marked in the article.

1) Bombyx mori and Eri should be written in Italic throughout the manuscript.

Replying:

Thanks for the reviewer’s kind suggestion.we have set the Bombyx Mori and Eri in the full text as italic.

2) In page 3, Figure 5 is first appeared in the manuscript. This is strange. Put Figures in an appropriate place in the manuscript.

Replying:

Thanks for the reviewer’s kind suggestion. We have adjusted Figure 5 to figure 3 and placed it in the appropriate place in the manuscript.

3) The font sizes in all Figures are not consistent. The authors should be more careful to prepare each Figure.

Thanks for the reviewer’s kind suggestion. We have increased the font size of figures 1-4 to ensure that the fonts of all figure are consistent.

4) In Figure 9, this reviewer cannot read the description due to the unclear letters in the Figure.

Thanks for the reviewer’s kind suggestion. We've adjusted the clarity of the figure and text.

5) Make clear what the purple color squares denote at the caption in Figure 4.

Thanks for the reviewer’s kind suggestion. The purple color square frame (bottom right) in figure.5 shows a partial enlarged view.

6) Figure 12, Y axis should not be “Weigh residue” but “Weight remaining”.

Thanks for the reviewer’s kind suggestion. We have replaced "weight residue" with "weight remaining" in Figure 9.

7) In line 25 in page 7, 1 um should be 1 µm.

Replying:

Thanks for the reviewer’s kind suggestion. We have replaced "um" with "µm" in Figure 9.

We tried our best to improve the manuscript and made some changes in the manuscript. These changes will not influence the content and framework of the paper. And the changes we have already marked in revised paper. We appreciate for Editors & Reviewers’ warm work earnestly again, and hope that the correction will meet with approval. Once again, thank you very much for your kind comments and suggestions.

Round 2

Reviewer 2 Report

Zhou, B. and Wang, H. investigated the structural, morphological, and mechanical properties of Eri silkworm. The manuscript is further improved after the revision with the intensive English editing. This reviewer thinks that the manuscript can be acceptable after minor revisions shown below:

  1. Throughout the manuscript, Eri silkworm cocoon is abbreviated as "E.cocoon" in italic. The word "cocoon" should not be italicized. The same is true for "B.cocoon".
  2. In Figure 12, Y axis should not be "Weigh remaining" but "Weight remaining" as suggested previously.

Author Response

Dear Reviewer,

Thank you for your kind suggestions and  comments for revising our paper (polymers-993884) titled “Structure and functions of cocoons constructed by Eri Silkworm” We have revised the manuscript accordingly, and all suggestions have been taken into full consideration. We sincerely hope our manuscript will be acceptable for publication in polymers.

We look forward to your response.

Sincerely yours,

Response to Reviewer’s Comments

1) Throughout the manuscript, Eri silkworm cocoon is abbreviated as "E.cocoon" in italic. The word "cocoon" should not be italicized. The same is true for "B.cocoon".

Replying:

Thanks for the reviewer’s kind suggestion. We have changed all the italicized words "cocoon" to regular font.

2) In Figure 12, Y axis should not be "Weigh remaining" but "Weight remaining" as suggested previously.

Thanks for the reviewer’s kind suggestion. We have replaced " Weigh remaining " with "weight remaining" in Figure 12.

We tried our best to improve the manuscript and made some changes in the manuscript. These changes will not influence the content and framework of the paper. And the changes we have already marked in revised paper. We appreciate for Reviewers’ warm work earnestly again, and hope that the correction will meet with approval. Once again, thank you very much for your kind comments and suggestions.
